# Urgency forces stimulus-driven action by overcoming cognitive control

Christian H Poth*

Neuro-Cognitive Psychology, Department of Psychology and Center for Cognitive Interaction Technology, Bielefeld University, Bielefeld, Germany

**Abstract** Intelligent behavior requires to act directed by goals despite competing action tendencies triggered by stimuli in the environment. For eye movements, it has recently been discovered that this ability is briefly reduced in urgent situations (Salinas et al., 2019). In a time-window before an urgent response, participants could not help but look at a suddenly appearing visual stimulus, even though their goal was to look away from it. Urgency seemed to provoke a new visual–oculomotor phenomenon: A period in which saccadic eye movements are dominated by external stimuli, and uncontrollable by current goals. This period was assumed to arise from brain mechanisms controlling eye movements and spatial attention, such as those of the frontal eye field. Here, we show that the phenomenon is more general than previously thought. We found that also in well-investigated manual tasks, urgency made goal-conflicting stimulus features dominate behavioral responses. This dominance of behavior followed established trial-to-trial signatures of cognitive control mechanisms that replicate across a variety of tasks. Thus together, these findings reveal that urgency temporarily forces stimulus-driven action by overcoming cognitive control in general, not only at brain mechanisms controlling eye movements.

*For correspondence:
c.poth@uni-bielefeld.de

Competing interest: The author declares that no competing interests exist.

## Editor's evaluation

It has been shown previously that saccades are obligatorily directed to visual stimuli if they are generated under time pressure, indicating that cognitive control is reduced briefly after a stimulus onset. The present study demonstrates this temporary impairment in cognitive control is present for manual responses, can occur when the conflict arises from non-spatial features of stimuli, and therefore is more general than previously thought.

## Introduction

Intelligent behavior requires that information about the environment is not only processed, but that it is reconciled with ongoing actions and momentary behavioral goals. This requires brain mechanisms for cognitive control, which enable to behave goal directed even if this means to overcome compelling tendencies for other behaviors (*Cohen, 2017*; *Egner and Hirsch, 2005*; *Gratton et al., 2017*). These brain mechanisms are assumed to be implemented in the frontal cortex (*Miller, 2000*; *Nee and D'Esposito, 2016*; *Ridderinkhof et al., 2004*; *Tang et al., 2016*), which is evident from the devastating impairments of goal-directed behavior in neurological disorders with frontal damages (*Alexander et al., 2007*; *Badre et al., 2009*; *Casey et al., 2002*). The brain mechanisms for cognitive control are thought to interface with attention mechanisms at multiple levels of processing (e.g., *Egner, 2008*), thereby enforcing that goal-relevant information is prioritized, whereas conflicting information is rejected (*Cohen, 2017*; *Gratton et al., 2017*).

*Salinas et al., 2019* found that the need for urgent responding opened up a time-window (~40 ms) in which the ability to perform goal-directed saccadic eye movements was severely reduced.

Participants had to look away from a suddenly appearing visual stimulus (an antisaccade), which requires to suppress the natural tendency to look toward it (*Munoz and Everling, 2004*). During the time-window, Salinas et al. observed that the eye movements were strongly drawn to the location of the visual stimulus, even though this conflicted with current task-goals. The authors interpreted this finding as evidence for an 'attentional vortex', a period in which the endogenous (goal-directed) control of spatial attention was overpowered by the exogenous attraction of spatial attention to the stimulus location. However, this interpretation in terms of attention control was hotly debated during eLife's open peer-review (https://elifesciences.org/articles/46359#SA1). Given the tight links of vision and eye movements, it was suggested that the findings reflected a specific visually driven impairment of eye movement control (oculomotor capture; e.g., *Irwin et al., 2000*), and that 'the link to attention is only inferred' (review round 2, point 1). Compatible with the suggestion, Salinas et al. modeled potential mechanisms underlying their findings based on neuronal activity of brain regions specialized in eye movement control (*Costello et al., 2013*; *Seideman et al., 2018*; *Shankar et al., 2011*; *Stanford et al., 2010*). Thus, in line with the suggestion, this modeling constrains interpretations to the domain of eye movements and closely related brain mechanisms of spatial attention (*Moore and Armstrong, 2003*).

The critical debate urges us to ask how general Salinas et al.'s discoveries are. Specifically, are the findings really limited to eye movement control and its linked attention mechanisms, or do they signify more wide-ranging influences on cognitive mechanisms? We show that the latter is the case. Going even beyond Salinas et al.'s 'attentional vortex', we found that urgency generally elicits stimulus-driven behavior that can conflict with current goals. In two well-established manual tasks of cognitive control, behavioral responses were dominated by task-irrelevant stimulus information. As a result, participants could not help but execute stimulus-driven responses that were in clear conflict with

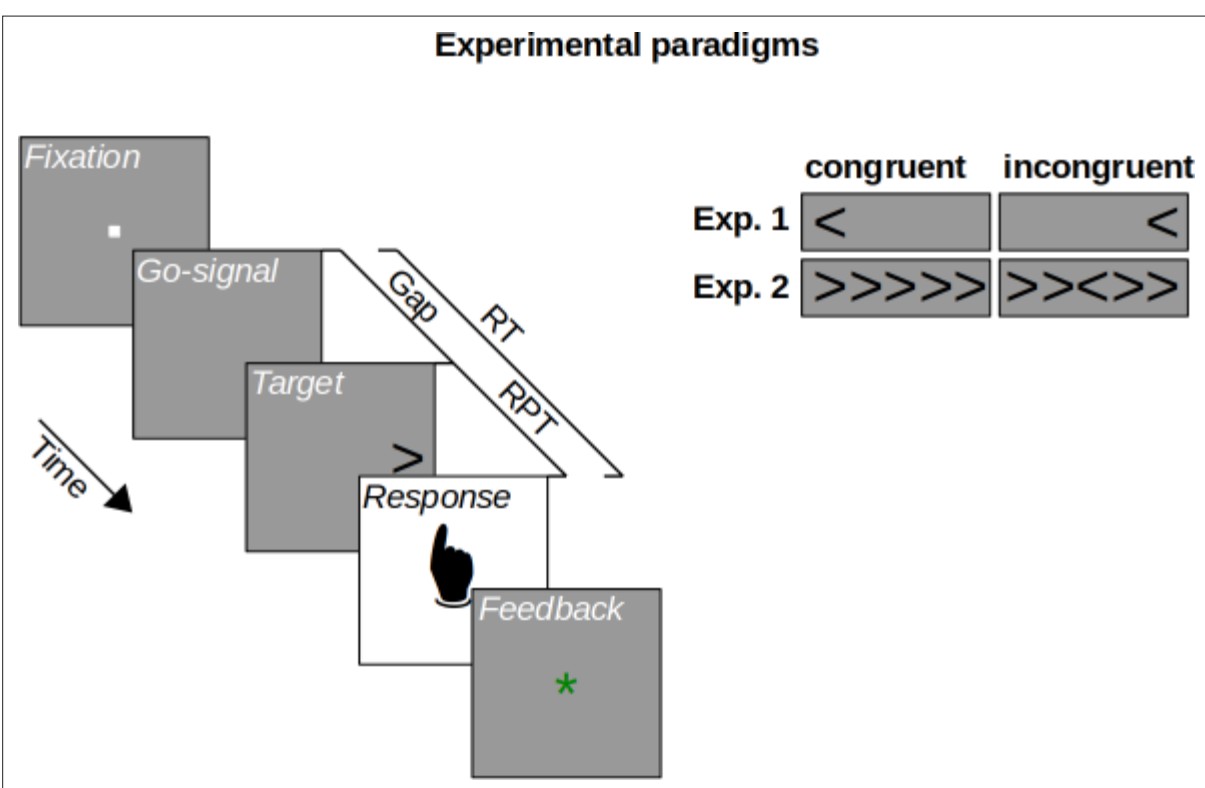

**Figure 1.** Experimental paradigms. On each trial, participants fixated a fixation stimulus whose disappearance served as a go-signal, prompting a response within a 1 s deadline. In both experiments, the target stimulus followed the go-signal after a variable gap duration. Shorter gaps imply lower urgency for responding, leaving enough time until the deadline. Longer gaps imply higher urgency for responding, leaving little or no time until the deadline. Participants responded by button press, receiving a feedback about whether they responded in time. Experiment 1 used a spatial Stroop task, in which participants indicated the direction of an arrow matching or mismatching (congruent vs. incongruent) its horizontal location. Experiment 2 used a flanker task, in which participants indicated the direction of a central arrow flanked by matching or mismatching (congruent vs. incongruent) distractor stimuli. Raw processing time (RPT) is the reaction time (RT) minus gap duration.

their current goals. Experiment 1 revealed such a dominance of stimulus-driven behavior for conflicts of spatial stimulus location, and Experiment 2 demonstrated the same pattern even for nonspatial cognitive conflicts. In both experiments, the dominance of stimulus-driven behavior followed trial-to-trial sequence effects that are interpreted as top-down and/or bottom-up modulations of cognitive control mechanisms (*Egner, 2017*; *Egner, 2007*). As such, these findings uncover that urgency temporarily helps external stimuli to overcome (e.g., bypass) the goal-driven cognitive control of action at processing levels extending beyond vision, spatial attention, and eye movement control (such as levels of prefrontal brain mechanisms; *Egner, 2007*; *Egner and Hirsch, 2005*).

## Results

### Urgency elicits a dominance of stimuli over cognitive control of spatial conflicts

In Experiment 1, cognitive control was assessed using a spatial Stroop task (*Clark and Brownell, 1975*; *Funes et al., 2010*; *Lu and Proctor, 1995*; *Schneider, 2020*). In this task, participants had to respond by pressing one of two buttons to indicate whether a target stimulus, an arrow (see *Figure 1*), was pointing left or right. The target arrow could appear either to the left or right of screen center. In the congruent condition, the target arrow appeared on the side it was pointing to. In the incongruent condition, the target arrow appeared on the side opposite to where it was pointing to. Such a condition has been shown to lead to slower reaction times, indicating that the spatial location and the pointing direction of the target arrow produced a cognitive conflict for action (e.g., *Lu and Proctor, 1995*; *Schneider, 2020*, Experiment 1). Cognitive control can be quantified by assessing how well this cognitive conflict can be resolved. This is done by measuring how performance suffers in the incongruent condition compared with the congruent condition. If cognitive control was perfect, performance should be equal in the conditions, and for worse cognitive control, performance should drop stronger in the incongruent compared with the congruent condition. Harnessing these classic congruency effects, we traced the evolution of cognitive control over time using a manipulation of urgency. The trial sequence is shown in *Figure 1*. In the beginning of a trial, participants fixated a small square in the center of the screen. The disappearance of this fixation stimulus served as the go-signal, prompting participants to respond after 1 s. Urgency was manipulated by introducing a gap of different durations after the go-signal and before the target stimulus that indicated the response to be made (*Salinas et al., 2019*; *Stanford et al., 2010*). When the duration of the gap is short, urgency should be low. The target appears just after the go-signal, allowing for processing the target in order to prepare the motor response. In contrast, when the gap duration is long, urgency should be high. During the gap after the go-signal, the time left for responding is elapsing, but the target specifying the response is still missing. Thus, time-pressure for responding is building up, so that participants must prepare their motor responses before it becomes clear which response has to be chosen. This manipulation may have induced urgency as an internal state of participants' cognitive system, as measures assumed to index physiological arousal were found to increase with increasing gap duration (*Appendix 1—figure 2*; but see the discussion there). Following *Salinas et al., 2019*, we computed the raw processing time (RPT) for the target as the time the target was visible within the reaction time (i.e., the reaction time from the go-signal minus the gap duration). For both conditions, we assessed the temporal evolution of performance by computing the tachometric function, which is the psychometric function relating performance to RPT (*Figure 2*). This revealed a qualitative difference between the congruent and incongruent condition (and closely replicated a pilot experiment, see *Appendix 1—figure 1*). In the congruent condition, performance rose monotonously from chance level up to an asymptote at near-perfect performance. In contrast, the tachometric function in the incongruent condition followed a qualitatively different course: While it also started at chance and evolved toward near-perfect performance, it dropped below chance level in a demarcated time-window around an RPT of about 250 ms. The maximum drop below chance was lower in the incongruent condition than the congruent condition ($p < 0.001$, permutation test, see Materials and methods; *Figure 2b*). This reveals that there was a time-window of RPTs in which responses were dominated by the spatial location of the target arrow rather than its pointing direction, even though this violated the task instructions. After this time-window, performance in the incongruent condition recovered up to near-perfect performance, but reached this level much later compared with the congruent condition (i.e., the tachometric function

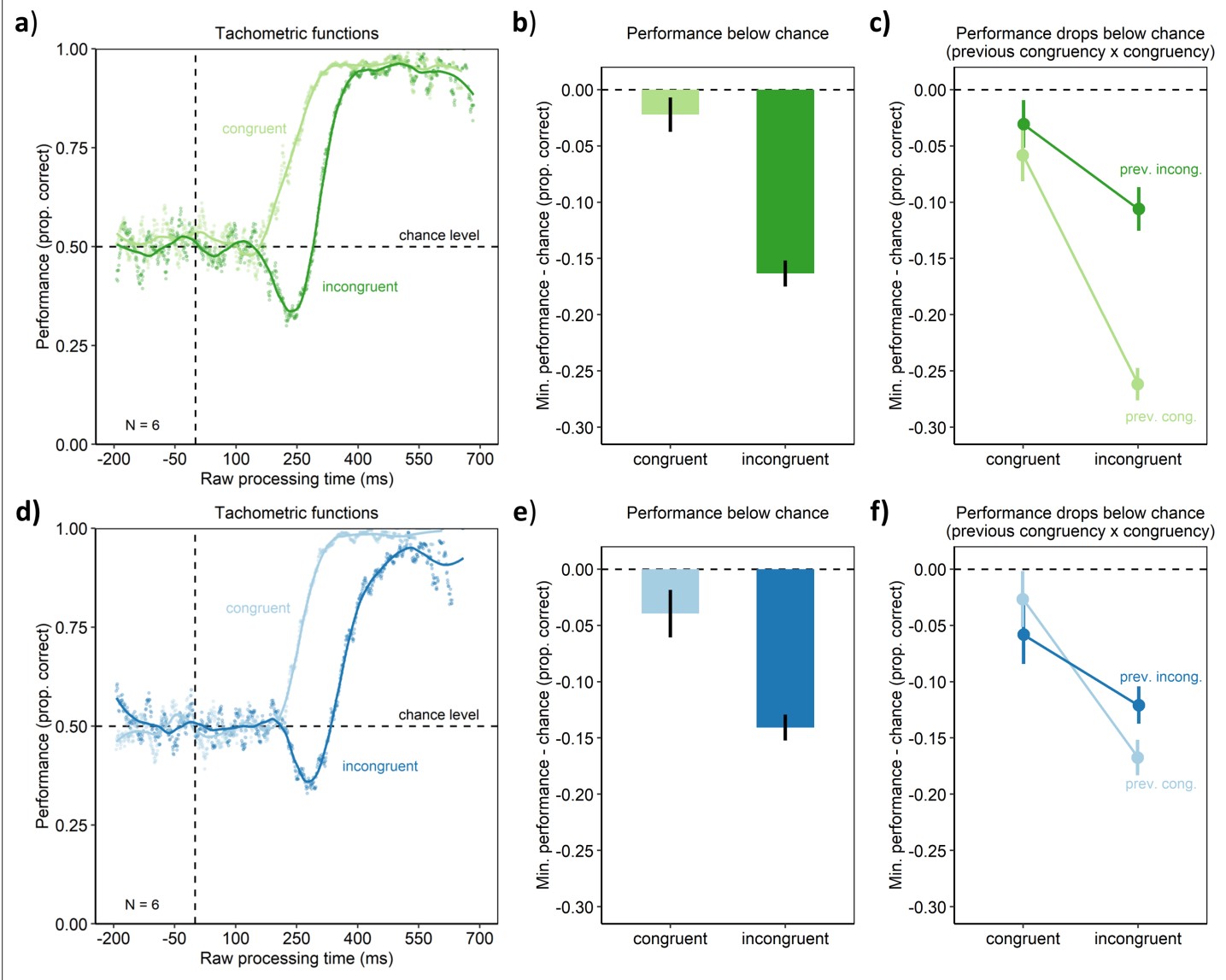

**Figure 2.** Behavioral results of Experiment 1 (upper panel) and Experiment 2 (lower panel). (**a, d**). Tachometric functions depicting performance as a function of RPT for the congruent and incongruent conditions of the aggregate participant. (**b, e**) In contrast to the congruent condition, the incongruent condition led to a larger drop of performance below chance. (**c, f**) The drop of performance below chance was larger, when the previous trial was congruent (prev. cong.) rather than incongruent (prev. incong.). Error bars are standard errors from a bootstrap.

was shifted to the right for ~80 ms, see *Figure 2a*). This indicates that cognitive control mechanisms eventually resolved the competition between stimulus information and current goals in favor of the goals, but that this resolution required a substantial amount of processing time.

One hallmark of cognitive control is that it is dynamic: Humans can adapt or learn to compensate cognitive conflicts (*Botvinick et al., 2001*; *Egner, 2007*; *Hommel et al., 2004*), so that performance decrements are less pronounced following trials with cognitive conflict (incongruent trials) than trials without such conflict. We discovered that this was also the case for our urgency-based effects: Performance dropped more strongly below chance level in the incongruent condition when the preceding trial was congruent rather than incongruent (p < 0.001, permutation test, *Figure 2c*). Thus, the drops of performance below chance exhibit the same trial-to-trial modulations of established congruency effects. This pattern of results offers converging evidence that indeed the established mechanisms of cognitive control are involved (e.g., *Egner, 2017*).

Taken together, the results of spatial Stroop task of Experiment 1 reveal a demarcated time-window along the urgency axis in which cognitive control is overcome, so that responses cannot be based on

the symbolic information of the target arrow (i.e., its pointing direction), but are involuntarily (against task instructions) based on the spatial location of the target. Salinas et al. observed similar effects for saccadic eye movements, in which urgency led to a time-window in which saccades were involuntarily drawn toward rather than against (as per the instructions) a suddenly appearing target. Thus, the present findings suggest that such a dominance of stimulus information over cognitive control is not limited to the capture of eye movements by sudden stimuli. Rather, cognitive control is overcome (e.g., bypassed) by the stimuli also in manual decision tasks. In Experiment 1, cognitive conflicts arose from the task-irrelevant spatial location of the target stimulus. Thus, the dominance over cognitive control could be limited to such spatial cognitive conflicts. Therefore, we performed Experiment 2 to test whether our present findings translated to nonspatial types of cognitive conflict.

## Urgency elicits a dominance of stimuli over cognitive control of nonspatial conflicts

In Experiment 2, cognitive control was assessed using a flanker task (*Eriksen and Eriksen, 1974*; *Ridderinkhof et al., 2021*). The paradigm and trial sequence were similar to the one of Experiment 1. However, here the target arrow always appeared at screen center. The target was flanked by four distractor arrows. In the congruent condition, the distractors pointed in the same direction as the target arrow, so that there was no cognitive conflict. In contrast, in the incongruent condition, the distractors pointed into the opposite direction as the target, thus providing conflicting information as to which response had to be executed. In contrast to Experiment 1, as the target always appeared at screen center, the conflict was not spatial in the sense that it was due to the target's spatial location. Results revealed the same pattern as in Experiment 1. Performance monotonously rose from chance to near-perfect performance with increasing RPT in the congruent condition (*Figure 2d*). In contrast, within a demarcated time-window of RPTs, performance dropped below chance level in the incongruent condition (p < 0.001, permutation test). Thus, even though the type of cognitive conflict was nonspatial, urgency resulted in the temporary dominance of responses by stimulus information from the distractors over goal-driven cognitive control. As in Experiment 1, after the time-window of this performance drop, performance in the incongruent condition recovered, but this required additional processing time (i.e., the tachometric function was shifted to the right for ~120 ms, see *Figure 2d*). Thus, again, cognitive control mechanisms eventually enacted goal-driven responses rather than stimulus-driven ones, but for this the mechanisms needed a substantial amount of processing time.

As in Experiment 1, this dominance of stimulus information was subject to the trial-to-trial sequence effects typical for cognitive control. That is, the performance fell below chance more strongly after trials with no conflict (congruent trials) compared to trials with conflict (incongruent trials; p = 0.019, permutation test, *Figure 2f*). Taken together, the results of Experiment 2 suggest that urgency elicits a circumscribed time-window in which stimulus-driven behavior overcomes cognitive control also for nonspatial cognitive conflicts.

## Discussion

Urgency elicited a time-window in which the capability to resolve cognitive conflicts between current goals and stimulus information was lost. During this time-window, participants' responses were dominated by stimulus information, forcing them to execute stimulus-driven responses against their current task-goals. This pattern of results closely resembles the effects of abruptly appearing visual stimuli on saccadic eye movements found by *Salinas et al., 2019*. However, the present findings reveal that these urgency effects have more wide-ranging implications than previously thought. Rather than causing a limited visual–oculomotor phenomenon, urgency seems to force stimulus-driven behavior overcoming cognitive control more generally.

Stimulus-driven behavior was temporarily enforced in two manual tasks, showing that this phenomenon is not restricted to saccadic eye movements. Compared with eye movements, this effect appeared for longer RPTs, in line with the generally longer reaction times for manual movements compared with eye movements. We found the phenomenon for cognitive conflicts in which the spatial location of stimuli conflicted with their task-relevant features (Experiment 1). This is in line with Salinas et al.'s interpretation that urgency caused an irresistibly strong capture of saccadic eye movements by the visual stimulus (oculomotor capture; *Theeuwes, 2014*) in their antisaccade

task. Given the tight links of eye movement control and spatial attention (*Deubel and Schneider, 1996*; *Moore and Armstrong, 2003*; *Rolfs et al., 2011*), such a capture should not only attract saccades (overt attention), but also covert spatial attention (without saccadic eye movements) prioritizing the target's location for perception (*Carrasco, 2011*; *Theeuwes, 2014*). For Experiment 1, such a prioritization of stimulus location could have provoked manual responses that were compatible in their spatial organization (*Lu and Proctor, 1995*). This would explain the findings of Experiment 1 by assuming mechanisms similar to those implicated in Salinas et al.'s eye movement paradigm. Critically, however, in Experiment 2 we found the dominance of stimulus-driven behavior also for cognitive conflicts that were nonspatial. Here, urgency elicited a time-window in which distractor stimuli flanking a target stimulus involuntarily dominated participants' responses. In both experiments, cognitive conflicts were eventually resolved, but this took a substantial amount of additional processing time (see *Salinas et al., 2019* for comparison). Thus, this underscores that action control happens faster if it is driven by stimulus information rather than cognitive control mechanisms enacting current goals.

Furthermore, in both experiments, the dominance of stimulus information over cognitive control was stronger after trials without conflict than trials with conflict. The precise nature of top-down and bottom-up influences underlying such trial-to-trial modulations is still debated (*Egner, 2017*). However, for our present view this pattern offers corroborating evidence that indeed those established cognitive control mechanisms were involved that are sensitive to these modulations (e.g., *Egner, 2017*). Thus, taken together, the present findings reveal that urgency forces stimulus-driven behavior by overcoming cognitive control not only in saccadic eye movements and spatial attention, but in goal-directed behavior more generally.

How could urgency induce such a temporary dominance of stimulus information over cognitive control? In an urgent task, the motor plans for the response alternatives are active after the go-signal for responding, even if the stimulus indicating which response should be made has not yet appeared (*Salinas et al., 2019*; *Stanford et al., 2010*). Thus, the motor plans are both active and compete, and this competition must be resolved 'on-the-fly' using incoming visual information. This can be achieved by pausing or accelerating the motor plans (*Salinas et al., 2019*; *Salinas and Stanford, 2018*) based on either the most salient stimulus information (in Experiment 1, the location of an abrupt onset, in Experiment 2 the information provided by the majority of the stimuli) or based on goal-relevant stimulus information, by means of cognitive control mechanisms. Now, in urgent situations, salient stimulus information could have privileged access to the motor plans because the cognitive control mechanisms are transiently interrupted that would otherwise filter out the information (and prevent it from biasing the motor plans). This may be part of an adaptive mechanism that allows salient stimuli to directly affect ongoing motor plans to enable that responses are delivered promptly (*Salinas and Stanford, 2018*; *Stanford and Salinas, 2021*).

We found that the time-pressure to respond (the gap duration), and thus urgency, was associated with two indices of physiological arousal, namely pupil size (*Mathôt, 2018*) and saccade peak velocity (*Di Stasi et al., 2013*). Although this association should be investigated further (for pupil size, there might have been influences of the stimulus sequence, see Appendix 1), these findings raise the hypothesis that urgency induces a momentary state of increased neuronal arousal (e.g., by modulating norepinephrine levels, *Aston-Jones and Cohen, 2005*). This would be similar to temporary states of phasic alertness, which occur after warning signals (alerting cues), and speed up cognitive information processing (*Bundesen et al., 2015*; *Petersen et al., 2017*; *Poth, 2019*). Thus, this would provide a way by which urgency could shorten the time over which the competition between motor plans extends. This could cause the salient information with its stronger neuronal representations to almost immediately decide the competition in its favor. No time would be left for cognitive control mechanisms to counteract the influence of this information (irrespective of whether cognitive control was transiently interrupted). This would be in line with the physiologically inspired computational model by *Salinas et al., 2019*, which explains the dominance of stimulus information over the cognitive control of saccades by a period in which faster exogenous signals already affect motor plans while slower endogenous signals have not yet been processed up to this level. This means that the dominance of stimulus information over cognitive control does not indicate that cognitive control mechanisms fail. Rather, it suggests that salient stimuli are used for action control before cognitive control mechanisms can exert their goal-driven influence (cf. *Salinas et al., 2019*).

It has long been established that cognitive conflicts slow down responses in a variety of cognitive tasks (*Lu and Proctor, 1995*; *Ridderinkhof et al., 2021*). However, one may wonder whether this actually makes a difference for behavior outside the laboratory, in real life. In most situations, it may have little consequences if a cognitive conflict delays a goal-directed response for a couple of tens (or even hundreds) of milliseconds. However, the present findings suggest that urgency may carry a danger here. Rather than just delaying goal-directed behavior, it enforces stimulus-driven behaviors that are adversarial to the current goals. The resulting behavioral errors may be a source of accidents in a number of safety-critical situations (e.g., in traffic).

## Conclusions

Urgency temporarily reduced the capability to handle cognitive conflicts. Behavior was involuntarily dominated by stimulus information, even though this information was adversarial to current goals. This dominance of stimulus information over cognitive control seems general in the sense that it (1) extends beyond previous reports of impaired eye movement control (*Salinas et al., 2019*), (2) occurs also in manual tasks, and (3) happens irrespective of whether cognitive conflicts arise from spatial stimulus locations or nonspatial stimulus information. Going beyond previous suspicions (*Salinas et al., 2019*), urgency seems to elicit a general 'attentional vortex in cognitive control', during which behavior is controlled by the external world rather than internal goals.

# Materials and methods

## Participants

Six participants (aged 21–23 years) performed Experiment 1, and six (aged 22–25 years) performed Experiment 2. These sample sizes were chosen in advance, based on *Salinas et al., 2019* and based on the pilot experiment that was replicated by Experiment 1 (see Appendix 1). They all reported (corrected-to-) normal vision and gave written informed consent before participating. The experiment followed the ethical regulations of the German Psychological Society (DGPs) and was approved by Bielefeld University's ethics committee.

## Experimental setup and tasks

Participants performed the experiments in a dimly lit room. They viewed the preheated (parameters as in *Poth and Horstmann, 2017*) computer screen (G90fB, ViewSonic, Brea, CA, USA) from a distance of 71 cm, while their right eye was tracked (Eyelink 1000, tower-mounted eye tracker, SR Research, Ottawa, ON, Canada). They responded using a standard computer mouse. The experiments were programmed using the Psychophysics Toolbox 3 (3.0.12; *Brainard, 1997*; *Kleiner et al., 2007*) and Eyelink toolbox (*Cornelissen et al., 2002*) for MATLAB (R2014b; The MathWorks, Natick, MA).

*Figure 1* illustrates an experimental trial. Participants fixated a central fixation stimulus (a 0.2 × 0.2° square, 85 cd/m²; against a background of 43 cd/m²) for 350, 400, or 500 ms, whose subsequent disappearance prompted them to respond within 1 s, even if the target stimulus for the response had not yet been shown. After a variable gap duration (0, 100, 200, …, 900, or 950 ms), the target stimulus appeared. Participants responded by pressing the left or right mouse button with their left or right index fingers, respectively. If they had responded within the deadline, they received a central green '*' (font size 20 px, 0.56°; 38 cd/m²) and otherwise a red '!' (font size 20 px, 0.56°; 22 cd/m²) as feedback (for 750 ms).

The participants' task was to indicate whether a target stimulus, a black arrow (*Figure 1*, font size 20 px, 0.56°, <1 cd/m²) pointed to the left or right. Participants of Experiment 1 performed a spatial Stroop task: The target appeared left or right (6°) to screen center. In the congruent condition, the target appeared on the side of its pointing direction, in the incongruent condition, it appeared on the other side. Participants of Experiment 2 performed a flanker task: The target appeared at screen center and was flanked left and right by two distractor arrows each. In the congruent condition, the distractors pointed in the same direction as the target, in the incongruent condition, in the other direction. In both experiments, participants were instructed to focus only on the target's pointing direction and ignore its spatial location or the distractors, respectively. In addition, they were instructed to prioritize responding within the deadline, even if it meant to guess their response.

Every participant performed 1188 experimental trials in five sessions, yielding 5940 trials in total. All combinations of the fixation durations, gap durations, pointing directions of the target, and congruency conditions occurred equally often and in randomized order within nine blocks per session.

## Data analysis

Data and analysis code are available on the Open Science Framework (https://osf.io/5rk6n/). Data were analyzed using R (4.1.0, https://www.R-project.org/). For each trial, the RPT was computed as reaction time minus gap duration (*Salinas et al., 2019*). Tachometric functions were derived by sliding a 1 ms bin across RPTs from −200 to 1000 ms and computing the average performance within the bin (*Figure 2a, d*). Trials with RPTs outside this range were excluded from analysis (Experiment 1: 9.56 %; Experiment 2: 21.19%). Performance (the proportion of correct trials) was then locally regressed on the RPT bins (using R's loess function with a span parameter of 0.2) to obtain separate tachometric functions for the congruent and incongruent condition. This was done to analyze the congruent and incongruent conditions with the same function, even though they had qualitatively different shapes (*Figure 2a, d*; this step was different from Salinas et al.'s approach that focused specifically on the shape of the incongruent condition).

As the data were consistent across participants, data analysis was performed on the data pooled across participants (following Salinas et al.'s analysis of their aggregate participant).

Experimental conditions were compared using permutation tests as follows. Original effects (e.g., differences between the congruent and incongruent condition in the minima of their estimated tachometric functions, see *Figure 2b,d*) were located in a distribution of effects from a 2000-fold reanalysis of the raw data with randomized condition labels, that is, trials labeled as congruent or incongruent.

## Acknowledgements

I thank Anika Krause for help with the data collection of Experiments 1 and 2, Niklas Dietze and Lukas Recker for comments on an earlier draft of this manuscript, and the team of the Neuro-Cognitive Psychology Group at Bielefeld University for helpful discussions. I acknowledge support for the publication costs by the Open Access Publication Fund of Bielefeld University.

## Additional information

### Funding

| Funder | Grant reference number | Author |
|---|---|---|
| Deutsche Forschungsgemeinschaft | 429119715 | Christian H Poth |

The funders had no role in study design, data collection and interpretation, or the decision to submit the work for publication.

### Author contributions

Christian H Poth, Conceptualization, Data curation, Formal analysis, Funding acquisition, Investigation, Methodology, Project administration, Software, Visualization, Writing – original draft, Writing – review and editing

### Author ORCIDs

Christian H Poth http://orcid.org/0000-0003-1621-4911

### Ethics

Human subjects: Participants gave written informed consent before participating. The experiment followed the ethical regulations of the German Psychological Society (DGPs) and was approved by Bielefeld University's ethics committee.

### Decision letter and Author response

Decision letter https://doi.org/10.7554/eLife.73682.sa1
Author response https://doi.org/10.7554/eLife.73682.sa2

## Additional files

### Supplementary files
• Transparent reporting form

### Data availability
Data and analysis code are available on the Open Science Framework (https://osf.io/5rk6n/).

The following dataset was generated:

| Author(s) | Year | Dataset title | Dataset URL | Database and Identifier |
|---|---|---|---|---|
| Poth CH | 2021 | Urgent cognitive control 001 | https://doi.org/10.17605/OSF.IO/5RK6N | Open Science Framework, 10.17605/OSF.IO/5RK6N |

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

## Appendix 1

### Pilot experiment

#### Results

Similar to Experiment 1, the pilot experiment assessed cognitive control using a variant of the spatial Stroop task (e.g., *Lu and Proctor, 1995*; *Schneider, 2020*). The experiment was similar to Experiment 1, but differed from it in a number of method details (see the methods below). The results of the pilot experiment are visualized in *Appendix 1—figure 1*. Tachometric functions were obtained and analyzed as for Experiment 1 (*Appendix 1—figure 1a*). As in Experiment 1, performance rose monotonously toward ceiling performance with increasing RPT in the congruent condition. In contrast, performance dropped below chance within a constrained time-window in the incongruent condition. This drop in the incongruent condition was statistically reliable, as indicated by a significantly lower minimum of the tachometric curve in the incongruent compared with the congruent condition (permutation test, p < 0.001, see *Appendix 1—figure 1b*).

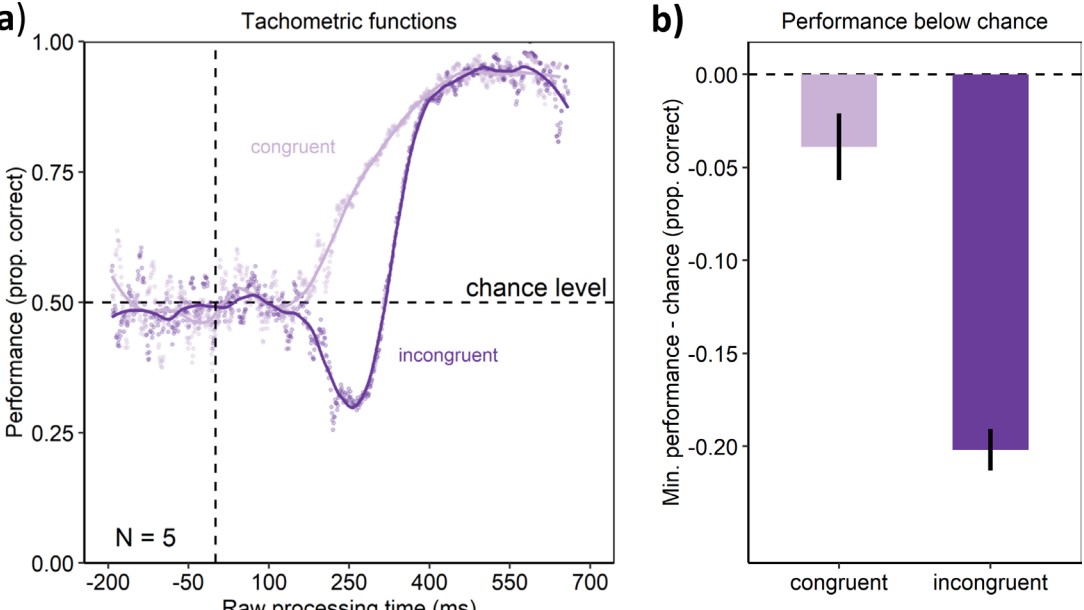

**Appendix 1—figure 1.** Behavioral results of the pilot experiment. (**a**) Tachometric functions depicting performance as a function of RPT for the congruent and incongruent conditions of the aggregate participant. (**b**) In contrast to the congruent condition, the incongruent condition led to a larger drop of performance below chance. Error bars are standard errors from a bootstrap.

#### Materials and methods

##### Participants

Five participants (aged 20–29 years) took part in the pilot experiment (one additional participant did not perform the experiment, because the experiment did not run on their home PC, on which the experiment was performed, see the setup below). They all reported (corrected-to-) normal vision and gave written informed consent before participating. The experiment followed the ethical regulations of the German Psychological Society (DGPs) and was approved by Bielefeld University's ethics committee.

##### Experimental setups and task

Participants performed the experiment at their homes using their own PCs (because laboratories were closed due to the COVID-19 crisis). They were instructed to set their screens at a refresh rate of 60 Hz (one participant's screen had a variable refresh rate for technical reasons, with about 75 Hz on average) and to view them from a distance of 71 cm. Responses were collected using external computer mouses. The experiment was programmed using PsychoPy2 (v1.85.3; *Peirce, 2008*; *Peirce, 2007*).

The task and the trial sequence was identical to the one of Experiment 1, with a few exceptions. Stimuli were shown against a black background. Target stimuli consisted of the letters 'L' or 'R' (Arial font, 1.5° height) on a white square (3.5 × 3.5°), appearing to the left or right (8° from screen center). The fixation stimulus was a small white square (0.1 × 0.1°). Participants performed 10 sessions of 568 trials each, in which all combinations of target letter (L vs. R), target location (left vs. right), fixation durations (350, 400, and 500 ms), and gap durations (0, 100, 200, 300, 400, 500, 600, 700, 800, 900, and 950 ms) occurred equally often.

## Data analysis

The data analysis was identical to the corresponding analyses of Experiment 1.

## Physiological arousal in Experiments 1 and 2

The momentary pupil size provides an often used measure of physiological arousal (e.g., *Mathôt, 2018*). For instance, pupil size has been taken as an index of phasic alertness, the arousal-driven increase in the brain's readiness for perception that follows warning signals ('alerting cues'; *Petersen et al., 2017*). The peak velocity of saccadic eye movements provides an index of physiological arousal as well (e.g., *Di Stasi et al., 2013*).

In Experiments 1 and 2, we recorded pupil size and saccade peak velocity as indices of physiological arousal. Specifically, we measured the average pupil size within 1 s after the manual response (*z*-scored for the participant's session) and the velocity of the first saccade (relative to its amplitude, *z*-scored for the participant's session, and detected using the Eyelink 1000 algorithm using a velocity threshold of 35°/s and an acceleration threshold of 9500°/s$^2$) after target onset.

As can be seen in *Appendix 1—figure 2*, for both experiments, and for both, the congruent and incongruent condition, the two arousal measures monotonously increased with increasing gap duration. This offers a first hint that introducing time-pressure for responding by this gap duration could change participants' momentary state of arousal. However, since the gap duration also marked the time after the offset of the fixation stimulus, the time-course of the visual stimuli might have contributed to the relation between gap duration and pupil size. In contrast to the pupil size, this explanation seems less likely for the peak velocity of the first saccade after target onset. After its offset, the representations of the fixation stimulus in early sensory memory (*Averbach and Coriell, 1961*; *Coltheart, 1980*) should have decayed within about 500 ms, and should thus have ceased to influence participants' saccades. As evident from *Appendix 1—figure 2*, however, participants' saccade peak velocities still continued to increase with increasing gap duration, also for the gap durations longer than 500 ms. Therefore, these data seem consistent with the assumption that the time-pressure induced by the longer gap durations led to an increase in participants' momentary arousal.

In sum, although further studies are required to settle the issue ultimately, the presented findings raise the new hypothesis that urgency increases physiological arousal. Interestingly, if this hypothesis was true, it would implicate urgency as a specific internal state. This state should affect the overall responsiveness of the perceptual and cognitive brain systems, for example by modulating overall neuronal activity (e.g., by modulating the norepinephrine system, *Aston-Jones and Cohen, 2005*). As such, urgency could operate similarly to phasic alertness, the temporary increases in responsiveness that for example happen after warning signals (e.g., *Petersen et al., 2017*; *Poth, 2019*).

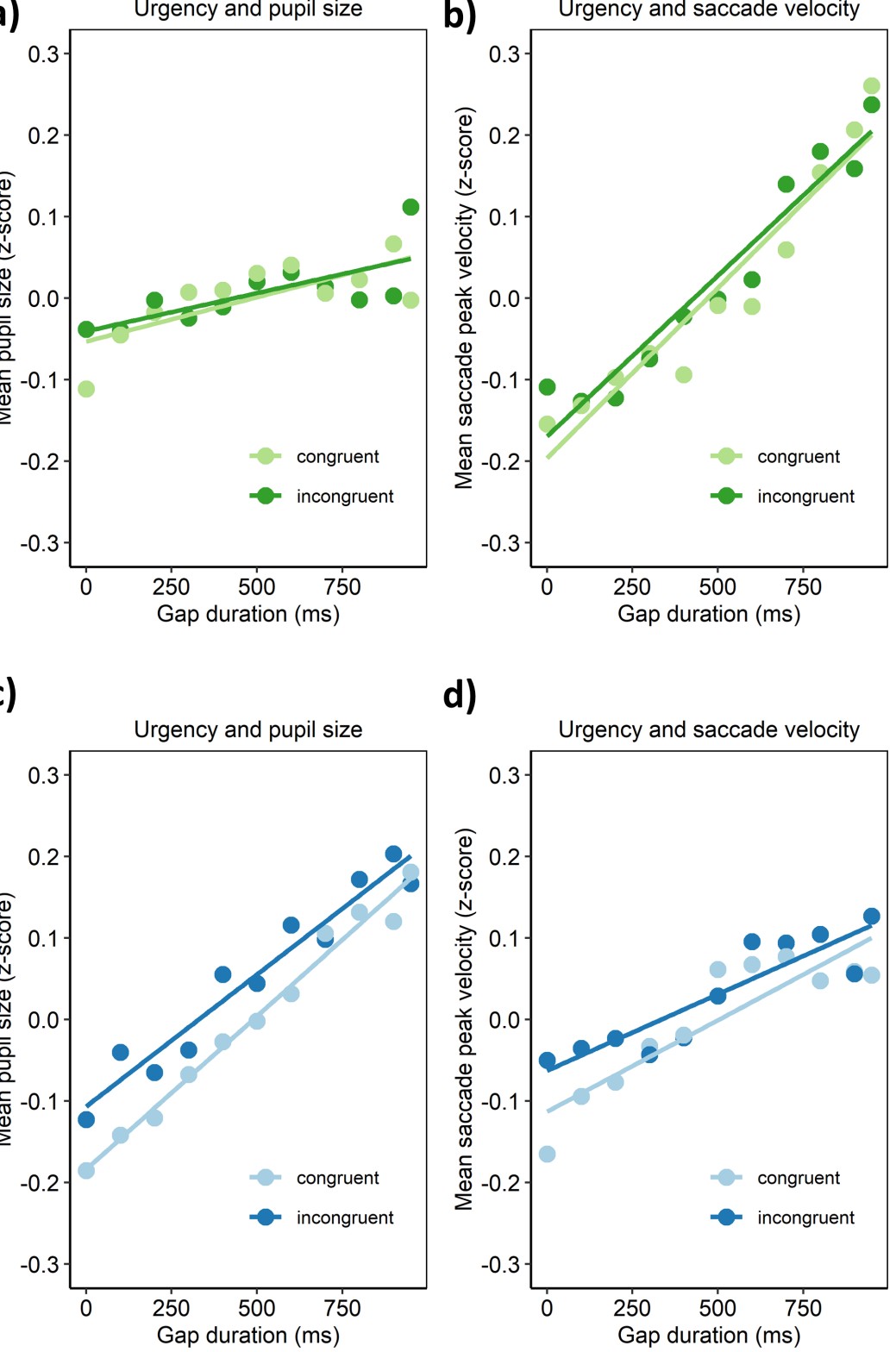

**Appendix 1—figure 2.** Relationships between urgency and measures of arousal in Experiment 1 (upper panel) and Experiment 2 (lower panel). (**a, c**) Urgency (gap duration) was associated with pupil size in both congruency conditions (average pupil size was assessed over 1 s after manual responses and z-scored across the trials of each session, see methods). (**b, d**) Likewise, urgency was associated with the peak velocity (relative to saccade amplitude) of the first saccade after target onset in both congruency conditions (see methods for saccade detection criteria).

