## [Editor Report]

It has been shown previously that saccades are obligatorily directed to visual stimuli if they are generated under time pressure, indicating that cognitive control is reduced briefly after a stimulus onset. The present study demonstrates this temporary impairment in cognitive control is present for manual responses, can occur when the conflict arises from non-spatial features of stimuli, and therefore is more general than previously thought.

---

## [Decision Letter]

**Decision letter after peer review:**

Thank you for submitting your article "Urgency disrupts cognitive control of human action" for consideration by *eLife*. Your article has been reviewed by 2 peer reviewers, including Daeyeol Lee as Reviewing Editor and Reviewer #1, and the evaluation has been overseen by Timothy Behrens as the Senior Editor. The following individual involved in review of your submission has agreed to reveal their identity: Emilio Salinas (Reviewer #2).

Essential revisions:

(1) The sequential effect of previous trial type (congruent vs. incongruent) appears to be robust in Experiment 1, and shows the same trend in Experiment 2. However, the effect seems smaller, and no information about the statistical test was provided. Was the effect still statistically significant?

(2) Interpretation of the relationship between pupil diameter (saccade peak velocity) and gap duration is problematic, because gap duration is also the time after fixation offset. Pupil diameter and saccade velocity might be affected directly by physical changes in visual stimuli, rather than simply reflecting the level of arousal.

(3) The language used to articulate the conclusion (as briefly in the public review) can be clarified. The wording includes phrases like "urgency disrupts cognitive control", "urgency temporarily impairs cognitive control", or "cognitive control collapses", which I think are a bit imprecise, strictly speaking, and have some unwarranted negative connotations. They make you think that something went wrong; that urgency *makes* our dear cognitive mechanisms fail.

First, such wording suggests that urgency is all that is needed for cognitive control to be turned off, but that is not quite correct. Urgency *enables* certain stimuli to acquire direct access to motor circuits; it allows those stimuli to bypass the normal filtering checkpoints. But it does not, in and of itself, turn those filters off. It is possible to set up urgent tasks similar to those reported here where the capture does not take place. The particulars of the visual input matter a lot.

Second, I don't think this is at all a failure, but simply part of the design of the system. My lab's working hypothesis about this is that what we're seeing is a mechanism that, when confronted with a new, unexpected, salient stimulus, prioritizes its analysis and interrupts any ongoing motor plans to determine if such stimulus is something that requires an immediate response. This is elaborated in a couple of publications (Salinas and Stanford, Sci Rep, 2018; Salinas and Stanford, Curr Opin Neurobiol, in press), but is consistent with a lot of phenomena related to stimulus detection (e.g., Bompas et al., Psychol Rev, 2020; Kozak and Corneil BD, J Neurophysiol, 2021).

As a comparison, one could state the conclusion as follows: under time pressure, certain features of (salient, abrupt onset) visual stimuli have privileged access to motor plans, and may bias responses without any regard to ongoing behavioral goals. This would not invoke any failures or collapses! Again, this is not a huge deal, but I think adjusting the language with these considerations in mind would provide a more accurate and nuanced conclusion as to the generality and significance of the results (by the way, the last paragraph, under "Conclusions", is much better at this than the beginning of the paper). It could also be an interesting discussion point, perhaps.

(4) The paper in general highlights the overt capture events (Figure 2b, e), but perhaps it is even more notable that the congruent and incongruent curves are shifted relative to each other by 50--100 ms, which is a lot of processing time. It is easy to envision conditions in which a shift is seen without a dip below chance. In any case, rather than a disruption or failure of cognitive control, such a shift is readily interpretable as a delay in cognitive control. The expected control does happen, but comes in later. This might be worth mentioning, especially given that the author acknowledges that cognitive control changes "temporarily", but does not really say what part of the data indicates that.

*Reviewer #2 (Recommendations for the authors):*

1. The data in figure S2 are nice. Perhaps consider promoting that figure and the few lines of text that go along with it to the main text.

2. Line 146: the analysis of history effects is another nice feature of this study. But again, I'm not sure that they are further evidence of a "collapse of cognitive control". To me, the presence of those effects suggests that the observed changes in performance are mediated by the very same attentional mechanisms that are normally present in standard, non-urgent tasks.

3. Figure 2: in panels a and d, I would suggest limiting the x axes to 700 ms, to better appreciate the scale of the relative shift.

Line 154: perhaps introduce parentheses to say "drawn towards rather than against (as per the instructions) a suddenly appearing target."

Line 203: should be "participants' responses".

Line 210: should be "this phenomenon is not".

Line 254: should be "if a cognitive conflict delays".

Line 320: should be "outside this range were excluded".

Line 328: should be "As the data were consistent".

---

## [Author Response]

Essential revisions:(1) The sequential effect of previous trial type (congruent vs. incongruent) appears to be robust in Experiment 1, and shows the same trend in Experiment 2. However, the effect seems smaller, and no information about the statistical test was provided. Was the effect still statistically significant?

I apologize for omitting this information. Yes, the effect was statistically significant and this is now described in lines 208 – 209.

(2) Interpretation of the relationship between pupil diameter (saccade peak velocity) and gap duration is problematic, because gap duration is also the time after fixation offset. Pupil diameter and saccade velocity might be affected directly by physical changes in visual stimuli, rather than simply reflecting the level of arousal.

I agree that we cannot rule out completely that the offset of the fixation stimulus has contributed to the increasing pupil size with increasing gap duration (because this offset led to a subtle darkening of the visual display). However, for the relationship between gap duration and saccade peak velocity, such an explanation seems less likely. After the offset of the fixation stimulus, the representations of this stimulus in early sensory memory (Averbach and Coriell, 1961; Coltheart, 1980) should have decayed quickly, within about 500 ms, and should thus have ceased to influence participants’ saccades. As evident from Figure S2 (now Appendix 1 - figure 2), however, participants’ saccade peak velocities still continued to increase with increasing gap duration, also for the gap durations longer than 500 ms. Therefore, these data seem more consistent with the assumption that the time-pressure induced by the longer gap durations led to an increase in participants’ momentary arousal. However, I agree that taken together the findings regarding the two arousal measures should be interpreted with caution.

Therefore, this point is now explicitly mentioned and discussed, both in Appendix 1 (p. 2, section “Physiological arousal in Experiment 1 and 2”) and in the main text (lines 132 – 135 and lines 294-296). In addition, the findings are interpreted more cautiously (lines 296-298).

(3) The language used to articulate the conclusion (as briefly in the public review) can be clarified. The wording includes phrases like "urgency disrupts cognitive control", "urgency temporarily impairs cognitive control", or "cognitive control collapses", which I think are a bit imprecise, strictly speaking, and have some unwarranted negative connotations. They make you think that something went wrong; that urgency makes our dear cognitive mechanisms fail.First, such wording suggests that urgency is all that is needed for cognitive control to be turned off, but that is not quite correct. Urgency enables certain stimuli to acquire direct access to motor circuits; it allows those stimuli to bypass the normal filtering checkpoints. But it does not, in and of itself, turn those filters off. It is possible to set up urgent tasks similar to those reported here where the capture does not take place. The particulars of the visual input matter a lot.Second, I don't think this is at all a failure, but simply part of the design of the system. My lab's working hypothesis about this is that what we're seeing is a mechanism that, when confronted with a new, unexpected, salient stimulus, prioritizes its analysis and interrupts any ongoing motor plans to determine if such stimulus is something that requires an immediate response. This is elaborated in a couple of publications (Salinas and Stanford, Sci Rep, 2018; Salinas and Stanford, Curr Opin Neurobiol, in press), but is consistent with a lot of phenomena related to stimulus detection (e.g., Bompas et al., Psychol Rev, 2020; Kozak and Corneil BD, J Neurophysiol, 2021).As a comparison, one could state the conclusion as follows: under time pressure, certain features of (salient, abrupt onset) visual stimuli have privileged access to motor plans, and may bias responses without any regard to ongoing behavioral goals. This would not invoke any failures or collapses! Again, this is not a huge deal, but I think adjusting the language with these considerations in mind would provide a more accurate and nuanced conclusion as to the generality and significance of the results (by the way, the last paragraph, under "Conclusions", is much better at this than the beginning of the paper). It could also be an interesting discussion point, perhaps.

I thank the reviewer for this helpful comment. I followed the suggestions to clarify the language in which the conclusions were presented throughout the manuscript.

Specifically, the following changes were made to the manuscript:

1. The title of the manuscript was changed from “Urgency disrupts cognitive control of human action” to “Urgency forces stimulus-driven action by overcoming cognitive control”.

2. The phrasings of the conclusions as “disruptions”, “impairments” or “failures” of cognitive control have been replaced. For instance, in the Abstract, the main conclusion is now phrased as:

“Thus together, these findings reveal that urgency temporarily forces stimulus-driven action by overcoming cognitive control in general, not only at brain mechanisms controlling eye movements.” (lines 48-50).

3. Throughout the whole manuscript, the conclusions are phrased as suggested. It is stressed that urgency should not “impair” the mechanisms for cognitive control, but instead should enable stimuli to overcome (e.g. bypass) these mechanisms and thereby obtain direct access to action control. For example, at the end of the Introduction, the main conclusion of the manuscript is previewed as:

“As such, these findings uncover that urgency temporarily helps external stimuli to overcome (e.g. bypass) the goal-driven cognitive control of action at processing levels extending beyond vision, spatial attention, and eye movement control (such as levels of prefrontal brain mechanisms; Egner, 2007; Egner and Hirsch, 2005)” (lines 99-103).

Likewise, throughout the Discussion, the conclusions have been rephrased in terms of a dominance of behavior by stimulus information rather than a failure of goal-driven cognitive control. Moreover, to avoid any ambiguity, it is explicitly stressed that “the dominance of stimulus information over cognitive control does not indicate that cognitive control mechanisms fail.” (lines 309-310).

4. The hypothesized mechanisms underlying the reported findings are now discussed in more detail (lines 276-290). Here, the suggested mechanistic interpretation of the findings from the public review is included as well (i.e. that “cognitive filtering mechanisms that normally mediate how we respond to visual stimuli are transiently interrupted under high urgency”, see below). That is, it is supposed that: “Now, in urgent situations, salient stimulus information could have privileged access to the motor plans because the cognitive control mechanisms are transiently interrupted that would otherwise filter out the information (and prevent it from biasing the motor plans)” (lines 285-290).

5. In addition, the interesting proposal that the findings could reflect an adaptive design feature of the cognitive system and not a failure of any cognitive mechanism is included in the Discussion as well:

“This may be part of an adaptive mechanism that allows salient stimuli to directly affect ongoing motor plans to enable that responses are delivered promptly (Salinas and Stanford, 2018; Stanford and Salinas, 2021).” (lines 287-290).

(4) The paper in general highlights the overt capture events (Figure 2b, e), but perhaps it is even more notable that the congruent and incongruent curves are shifted relative to each other by 50--100 ms, which is a lot of processing time. It is easy to envision conditions in which a shift is seen without a dip below chance. In any case, rather than a disruption or failure of cognitive control, such a shift is readily interpretable as a delay in cognitive control. The expected control does happen, but comes in later. This might be worth mentioning, especially given that the author acknowledges that cognitive control changes "temporarily", but does not really say what part of the data indicates that.

I agree and followed the suggestion. The shifts of the tachometric curves between the two conditions are now described with the results of Experiment 1 (lines 150-155) and Experiment 2 (lines 198-203). In addition, these findings are also included in the Discussion (lines 260-263).

Reviewer #2 (Recommendations for the authors):1. The data in figure S2 are nice. Perhaps consider promoting that figure and the few lines of text that go along with it to the main text.

I agree with this comment. However, because this data is not the main point of the paper and because it is now interpreted a bit more cautiously in response to Essential Revision (2) (see above), I now placed the figure in an appendix (Appendix 1).

2. Line 146: the analysis of history effects is another nice feature of this study. But again, I'm not sure that they are further evidence of a "collapse of cognitive control". To me, the presence of those effects suggests that the observed changes in performance are mediated by the very same attentional mechanisms that are normally present in standard, non-urgent tasks.

As described in the response to Essential Revision (3), the phrasing of the conclusions has been clarified throughout the text, so that interpretations as “collapses of cognitive control” are now avoided. Following the suggestion, the interpretation of the history effects is now phrased as:

“Thus, this pattern of results offers converging evidence that established mechanisms of cognitive control are involved (e.g., Egner, 2017).” (lines 164-166; see also lines 268-270).

3. Figure 2: in panels a and d, I would suggest limiting the x axes to 700 ms, to better appreciate the scale of the relative shift.

The figure was changed as suggested. For consistency, Figure 1 of Appendix 1 was changed as well.

Line 154: perhaps introduce parentheses to say "drawn towards rather than against (as per the instructions) a suddenly appearing target."

This was done (line 173).

Line 203: should be "participants' responses".

This was done (line 235).

Line 210: should be "this phenomenon is not".

This was done (line 244).

Line 254: should be "if a cognitive conflict delays".

This was done (line 317).

Line 320: should be "outside this range were excluded".

This was done (line 386).

Line 328: should be "As the data were consistent".

This was done (line 394).